# The Versatility of Collagen in Pharmacology: Targeting Collagen, Targeting with Collagen

**DOI:** 10.3390/ijms25126523

**Published:** 2024-06-13

**Authors:** Francisco Revert-Ros, Ignacio Ventura, Jesús A. Prieto-Ruiz, José Miguel Hernández-Andreu, Fernando Revert

**Affiliations:** Mitochondrial and Molecular Medicine Research Group, Facultad de Medicina y Ciencias de la Salud, Universidad Católica de Valencia San Vicente Mártir, 46001 Valencia, Spain; fj.revert@ucv.es (F.R.-R.); ignacio.ventura@ucv.es (I.V.); jesus.prieto@ucv.es (J.A.P.-R.); jmiguel.hernandez@ucv.es (J.M.H.-A.)

**Keywords:** collagen, drug, cancer, fibrosis

## Abstract

Collagen, a versatile family of proteins with 28 members and 44 genes, is pivotal in maintaining tissue integrity and function. It plays a crucial role in physiological processes like wound healing, hemostasis, and pathological conditions such as fibrosis and cancer. Collagen is a target in these processes. Direct methods for collagen modulation include enzymatic breakdown and molecular binding approaches. For instance, Clostridium histolyticum collagenase is effective in treating localized fibrosis. Polypeptides like collagen-binding domains offer promising avenues for tumor-specific immunotherapy and drug delivery. Indirect targeting of collagen involves regulating cellular processes essential for its synthesis and maturation, such as translation regulation and microRNA activity. Enzymes involved in collagen modification, such as prolyl-hydroxylases or lysyl-oxidases, are also indirect therapeutic targets. From another perspective, collagen is also a natural source of drugs. Enzymatic degradation of collagen generates bioactive fragments known as matrikines and matricryptins, which exhibit diverse pharmacological activities. Overall, collagen-derived peptides present significant therapeutic potential beyond tissue repair, offering various strategies for treating fibrosis, cancer, and genetic disorders. Continued research into specific collagen targeting and the application of collagen and its derivatives may lead to the development of novel treatments for a range of pathological conditions.

## 1. Introduction

Human collagens are oligomeric proteins that encompass a family of 28 members, 44 genes, and 1 pseudogene with 2 variants ([1] and Table 1). The basic molecular units, or protomers, of collagen consist of three twisted α chains, i.e., a triple-helical polypeptide. The steric restriction of the triple helix requires a primary structure with collagenous domains, repetitions of Gly-X-Y motifs, with X and Y frequently being proline and 4-hydroxyproline, which also stabilize the quaternary structure. In collagens I, II, III, and V/XI, 3-hydroxyproline has also been identified. The protomers are normally homotrimers, in which the three polypeptides are the same since most of the collagens have just one type of polypeptide. However, there are 7 types of collagen with more than one α chain (I, IV, V, VI, VIII, IX, XI). Among these, homotrimers have been observed only in collagen VIII, while the rest are heterotrimers of polypeptides of the same family. Just two mixed-heterotrimeric protomers have been described: α1(II)-α1(XI)-α2(XI) and α1(II)-α1(V)-α1(XI). Although the number of potential heterotrimeric combinations is very high, the number of characterized heterotrimers is quite limited ([1] and Table 1). Enclosed, please find a comprehensive list of collagen genes in the OMIM database (Appendix A).

The utilization of two alternative promoters in the *COL9A1* and *COL18A1* genes results in the production of distinct forms of the encoded chains [2,3]. Additionally, alternative splicing contributes to the presence of multiple isoforms in 10 collagen genes (Table 1). The process of alternative splicing is recognized for exhibiting tissue- or development-specific patterns, as exemplified by *COL2A1* [4]. However, in various instances, the functional consequences remain elusive. Notably, various types of collagens, such as IX (which binds to collagen II), XII, XIV, XV, and XVIII, are known to harbor glycosaminoglycan chains. These chains include chondroitin sulfate, dermatan sulfate, and heparan sulfate, which can be present in diverse combinations. This characteristic classifies them as proteoglycans [5,6,7,8,9].

The definition of collagen is blurry since there are a variety of quaternary structures, cellular and tissue localizations, and functions that are not necessarily related. Several molecules with collagen-like domains have not been classified as collagens. These molecules could be extracellular matrix components (e.g., emilins) playing a structural role [10]; however, in other cases, the collagen-like trimeric structure is part of the molecule that tightens its interactive and functional domains (e.g., adiponectin or complement 1q) [11].

The extracellular functions of collagen could be understood from a phylogenetic point of view. Collagen IV is the first ancestor in the phylogenetic tree, and it is directly associated with the transition to multicellularity by enabling the genesis of multicellular epithelial tissues [12]. It is noteworthy that some ancestors of collagen IV did not associate with epithelial differentiation [12,13]; in this sense, human collagen IV displays structural diversification [14,15,16], which could be associated with mesenchymal microenvironments [15]. The subsequent position on the phylogenetic tree of fibrillar collagens (collagen I) correlates with the structure of the mesenchyme and the connective tissues derived from it ([12] and Figure 1).

When viewed collectively as a family, the expression and distribution of collagen span a wide range. However, upon closer examination of each type of collagen individually, it becomes apparent that their biological functions are primarily tissue- or organ-specific. This observation is supported by the variety of diseases caused by pathological mutations identified to date, as outlined in Table 1 and Appendix A. 

## 2. Collagen-Related Diseases

Targeting collagen becomes logical when the focus is on diseases marked by excessive collagen production or where collagen plays a pivotal role in exacerbating pathological processes. On the other hand, a mutation in collagen genes could be the cause of a disease (Table 1 and Appendix A). In this case, the expression of recombinant or genome-edited collagens in the affected tissue could be an option. While the following section is not comprehensive, it offers a brief overview of the role of collagens in several pathophysiological conditions, focusing on diseases treated in the following sections.

### 2.1. Fibrosis and Wound Healing

Fibrosis is a pathological process characterized by the excessive accumulation of scar or fibrous tissue. Fibrotic tissue primarily consists of types I and III collagens, along with a combination of fibrotic cells [17]. Tissues displaying an excess of fibrosis often encounter prolonged healing challenges, potentially resulting in dysfunction of organs or tissues. It is also a pathogenic factor in cancer (see below) [18]. 

Fibrosis may occur either primarily, with no identifiable cause (idiopathic) or as a secondary response to a range of factors. These factors include chronic infections or inflammation, autoimmune diseases, exposure to toxins, radiation, or chemical compounds, as well as conditions like atherosclerosis, venous insufficiency, and cancer [17,19]. 

Depending on the organ affected by fibrosis, a number of diseases have been described. Idiopathic pulmonary fibrosis has been associated with alterations in the deposition of type I and type III collagen, which contribute to pulmonary interstitial remodeling and respiratory dysfunction [20]. Exposure to chemicals such as asbestos, cigarette smoke, drugs, or radiation or pathogens, can lead to secondary pulmonary fibrosis [21]. Other primary fibrotic phenomena include Dupuytren’s contracture, a chronic connective disorder of the palmar fascia of the hand, resulting in a progressive contracture in flexion of the finger [22]; and Peyronie’s disease, a condition in which men develop plaques of fibrous tissue in their penis, leading to pain or difficulty in sexual intercourse [23].

Liver fibrosis is normally secondary to chronic damage (alcohol, viruses, etc.). An increase in collagen type I deposition starting in the portal triads leads to fibrogenesis and the progression of hepatic cirrhosis. Stellate cells situated within the hepatic sinusoids play a pivotal role in the advancement of liver fibrosis [24]. A similar condition is found in the pancreas, where pancreatic stellate cells mediate fibrosis, potentially leading to chronic pancreatitis and cancer [25]. Cardiac fibrosis can also be secondary to various phenomena, resulting in distinct histological patterns. Ischemic cardiomyopathy, for instance, prompts interstitial fibrosis, which replaces the original muscle tissue. In contrast, dilated cardiomyopathy, which could be the end stage of hypertensive heart disease, induces diffuse fibrosis throughout the muscle tissue [26]. Although type I collagen traditionally takes center stage in these processes, recent findings have revealed that type IV collagen, along with other collagens, also significantly contributes to the scarring process in necrotic cardiac muscle [27].

Systemic sclerosis, also known as scleroderma, is a fibrotic condition that impacts various organs due to an autoimmune process. This chronic disease, whose etiology remains unknown, is typified by extensive fibrosis and vascular aberrancies occurring throughout the skin, joints, and internal organs, notably the esophagus, lower gastrointestinal tract, lungs, heart, and kidneys. Scleroderma (skin thickening) is indeed the expression of systemic sclerosis in the skin. In skin biopsy, atrophic areas are evident, showing increased collagen deposition and a reduction in the spaces between normal collagen bundles [28]. Keloids result from abnormal wound healing processes characterized by excessive accumulation of collagen within the scar tissue [29]. Renal sclerosis, distinct from fibrosis, is linked to alterations in the expression of type IV collagen. Changes in the deposition of collagen type IV play a role in glomerular dysfunction and the advancement of chronic kidney disease secondary to certain glomerulopathies, such as IgA nephropathy [30]. The ultimate stage would be fibrosis, marked by an early increase in the expression of collagen I and III [31].

### 2.2. Cancer

Network collagens and fibrillar (plus FACIT) collagens play different roles in the pathogenesis and progression of cancer. Their overexpression has been described in multiple types of cancer. However, both families of collagens interact with different integrins and discoidin domain receptors (DDRs) to initiate multiple signaling pathways that lead to cancer cell proliferation, survival, invasion, and metastasis [32,33]. 

The distribution of network collagens varies significantly. Epithelial cells are the main producers of collagen IV, while collagen VIII is predominantly located in corneal and vascular endothelial tissues, and collagen VI and X are typically found in connective tissues, where they are associated with fibrillar collagens. Therefore, tumors originating from epithelial cells are primarily associated with collagen IV expression in their pathogenesis. The progression of these tumors could be correlated with a shift in the type of collagen IV being secreted, which provides survival signals to cancerous cells that have acquired mesenchymal properties [15].

A relatively high expression of fibrillar collagen is associated with an unfavorable prognosis in many types of tumors, which is especially evident in papillary and clear-cell kidney cancers and bladder cancer. Fibrillar collagens also interact with integrins and DDRs to regulate the immune response [33]. Interestingly, while an excess of collagen I is considered pro-tumoral [34], the expression of collagen III is associated with a reduction of metastasis in murine models of breast cancer [35]. Moreover, collagen III induces quiescence in cancerous cells [36]. Nevertheless, it remains a negative prognostic indicator [33]. 

### 2.3. Hemostasis

Collagen plays a crucial role in blood clotting, particularly in the formation of the initial platelet plug. When blood vessels are damaged under flow, platelets adhere to the exposed collagen via the glycoprotein GPIb. Collagen does not bind directly to the platelet receptors but through a molecular bridge called von Willebrand factor (VWF). This initial adhesion triggers a series of events leading to platelet activation and aggregation, forming a platelet plug at the site of injury. Subsequent interaction of platelets with collagen, via GPVI or integrin α2β1, and with other ECM components, improves the stability of the plug. The integrity of the vessels depends on type I, III, IV, and VI collagens. Meanwhile, multiplexing found in the basement membrane mediates the inflammatory response [37,38]. 

Strategies to impede the functions of collagen that contribute to disease progression can be pursued through direct or indirect approaches (Figure 2). Indirect methods involve targeting proteins essential for collagen’s structural organization or its function within the disease context. Alternatively, therapeutic interventions may involve the utilization of collagens or derived peptides to activate biochemical processes. 

### 2.4. Genetic Disorders

Several diseases are caused by mutations in the genes encoding collagen or collagen-related proteins (Appendix A). The pathophysiology encompasses extracellular (reduced matrix) as well as intracellular effects (ER stress, apoptosis) [39].

Three pathological conditions have been treated using gene therapy procedures: osteogenesis imperfecta, which results from mutations in the *COL1A1* or *COL1A2* genes and can manifest with bone fragility, blue sclerae, hearing loss, and dental issues [40]; epidermolysis bullosa, a consequence of mutations in *COL7A1*, which is characterized by a fragile skin that blisters and tears from minor friction or trauma [41]; and junctional epidermolysis bullosa, with mutations in *COL17A1*, and similar symptoms [41].

## 3. Direct Targeting

### 3.1. Collagenases 

Numerous microorganisms secrete enzymes that break down collagen, facilitating the invasion of human tissues. The corresponding recombinant enzymes could be employed to digest collagen in vivo. One notable example is the collagenase from *Clostridium histolyticum* (CCH), which is effective and well tolerated in the short term in patients with Dupuytren’s contractures [42] and has been authorized for localized subcutaneous treatment. The recurrence of Dupuytren’s contracture treated with CCH is relatively low, demonstrating an efficacy similar to that of alternative treatments such as fasciectomy [43,44]. CCH is also one of the best treatments for Peyronie’s disease [45]. However, the CCH formulation known as Xiapex^®^ (or AA4500) is presently unauthorized by the European Medicines Agency. The withdrawal was at the request of the marketing authorization holder, Swedish Orphan Biovitrum AB (publ), which notified the European Commission of its decision to permanently discontinue the marketing of the product for commercial reasons [46].

While CCH has shown relative safety and efficacy, it can trigger the production of neutralizing antibodies in patients [47]. Additionally, AA4500 has proven ineffective in treating other fibrotic conditions, such as adhesive capsulitis of the shoulder [48]. Potential future applications abound. For instance, administering a collagenase solution within the abdominal cavity reduced tumor volume in animal models [49] and could potentially enhance chemotherapy effectiveness [50].

Another approved formulation of CCH is an ointment, marketed under the brand name Santyl^®^, which has received FDA approval for debriding chronic dermal ulcers and severely burned areas. The enzyme releases matrix fragments (mainly non-collagen peptides) that improve healing in vivo [51]. Interestingly, collagen degradation appears to be linked to an anti-inflammatory immune response and the production of TGFβ [52].

There are several eukaryotic enzymes responsible for the turnover of collagen in the extracellular matrix. Collagenases, or matrix metalloproteinases 1, 8, 13, and 18 (*MMP-1*, *MMP-8*, *MMP-13*, *MMP-18*), and gelatinase A and B (*MMP-2* and *MMP-9*, respectively) degrade essentially different types of collagens [53,54]. These enzymes are members of a large family of zinc-dependent extracellular matrix remodeling endopeptidases with additional substrates such as aggrecan, entactin, fibronectin, laminin, elastin, and other components of the extracellular matrix. Its enzymatic activity is regulated by natural inhibitors (tissue inhibitors of MMPs or TIMPs) [53,54]. MMPs and TIMPS are also involved in cancer and many other diseases [54]. In this case, a high or dysregulated activity had been associated with cancer progression. Moreover, several relevant pharmaceutical companies developed their own inhibitors (Table 46 from [55]), leading to a competitive race that resulted in a fiasco [56]. The first therapeutic approach was with pan-inhibitors of MMPs; however, no drug has been demonstrated to be effective for the treatment of cancer. Additionally, treatments were not well tolerated [56]. In this case, the effect on collagen is indirect. Recent efforts include specifically targeting MMP-9 with a monoclonal antibody (Andecaliximab/GS-5745), which resulted in unsatisfactory outcomes in gastric cancer [57,58], and Chron’s disease [59]. To our knowledge, MMPs have not been proposed as therapeutic agents for directly breaking down collagen in vivo.

### 3.2. Targeting Collagen with Biding Proteins and Peptides

Tumor collagen has been proposed as a specific target. Collagen-binding macromolecules could easily bind and accumulate in the tumor collagen. Consequently, targeting the collagen of the tumor extracellular matrix could enhance the accumulation and retention of immunotherapy drugs, thus improving their efficacy and reducing adverse effects. Collagen-binding domains (CBD) or proteins (CBP) have been proposed as candidates in tumor-targeting immunotherapy. CBD is a type of protein domain that binds specifically to collagen. CBD is derived or designed from collagen-binding sites in natural ligands of collagen, such as fibronectin (FN), VWF, placental growth factor (PlGF), and collagenase [60]. It has been reported that FN binds to types I and II collagens with high affinity [61,62]. Since FN plays an important role in wound healing and human growth, its collagen-binding domain has been mainly used to promote wound healing or treat developmental defects [63]. VWF, which mediates the adhesion of blood platelets, also promotes hemostasis by stabilizing coagulation factor VIII [64]. VWF could bind with high affinity to collagen types I, III, IV, and VI [65]. Some CBD of VWF has been used in collagen-targeting therapies for vascular repair [66], bone regeneration [60], and tumor treatment [67]. PlGF, a member of the VEGF family, participates in promoting angiogenesis, chondrogenesis, wound healing, and tumor growth [68]. Fragments of PlGF exhibit remarkably robust and indiscriminate binding to ECM components, not limited to collagen I. They have been utilized for delivering growth factors for tissue repair purposes [69]. Small collagen-binding peptides have been designed for better application with effective collagen-binding ability. Some examples are the heptapeptide TKKTLRT [70], used in regenerative medicine for efficient targeting of drugs to collagen, among other applications [71], and collagen mimetic peptides (CMP), especially interesting in diseases that present collagen degeneration [72].

In addition to collagen-binding domains, some small proteins such as lumican, bacterial surface proteins and avimers with collagen affinity are also assayed. Lumican is present in the ECM of cornea, gristle, and skin [73], and it has been shown to bind to collagenous fibers, strengthen their structure, and potentiate wound healing [74]. Lumican fused to therapeutic cytokines has been used for cancer immunotherapy, increasing the retention of drugs within tumors and reducing adverse effects [75]. Bacterial surface proteins such as lipoprotein SLR, M, and M-like proteins possess collagen affinity for infection at collagen-exposed wounds [76]. Finally, some avimers (a numerous group of small proteins that mediates the interaction between proteins) showing high collagen affinity were fused to IL-1 for the treatment of joint diseases [77].

## 4. Indirect Targeting

Given the intricate nature of the multiple stages associated with the development of a complex supramolecular structure, there exists considerable potential for identifying indirect targets within the process of collagen formation (Figure 2).

### 4.1. Translation

One of the most promising opportunities could arise at the translation level. The mammalian target of rapamycin complex 1 (mTORC1) phosphorylation of La-related protein 6 (*LARP6*) boosts the translation of a limited number of collagen mRNAs [78]. To date, the only described substrates of *LARP6* are *COL1A1*, *COL1A2*, and *COL3A1*. Furthermore, LARP6 appears to be dispensable for the baseline expression of these collagens, indicating that a targeted pharmacological intervention linked to its overexpression could be achieved, potentially reducing adverse reactions. In this sense, the first tricyclic compound displaying inhibitory activity of LARP6 (C9) shows promising prophylactic and therapeutic effects in several rat models of liver fibrosis. It was isolated through an in vitro screening process involving 50,000 drug-like compounds [79]. A broader pharmacological effect can be observed with the inhibition of mTOR (part of mTORC1 and mTORC2). Along with the downregulation of collagen expression, this pathway modifies the expression of additional genes involved in epithelial-mesenchymal transition, fibrosis, inflammation, survival, autophagy, and other pathways [80]. Silibinin, a natural flavonoid, which has been described as a potential STAT3 (Signal transducer and activator of transcription 3) inhibitor with antimetastatic activity [81], has been also identified as a suppressor of the mTOR signaling pathway, reducing collagen I and III expression [82]. However, at this moment, no results from animal models have been reported.

MicroRNAs (miRNAs) are small, non-coding RNA molecules that regulate gene expression by binding to target mRNAs, leading to their repression or degradation. They play a role in both physiological and pathological conditions [83]. The microRNA miR-29, with three mature members, miR-29a, miR-29b, and miR-29c, has been described as a mediator of the TGF-β pathway, inhibiting the expression of various collagens (I, III, IV, or XV) and other proteins in the extracellular matrix [84,85]. It plays a relevant role in fibrosis [86]. The pharmacological potential of miR-29a has been demonstrated in a mouse model of liver fibrosis induced by CCl_4_. Injection of miR-29a into the vein tail improved fibrosis by inhibiting the expression of *COL1A1* mRNA, among other possible mechanisms [87]. Its prophylactic and therapeutic effects in vivo have been recently confirmed using a bleomycin-induced lung fibrosis mouse model intravenously treated with a more stable mimic of miR-29 (MRG-229) [88]. MRG-229, an enhanced formulation of Remlarsen (MRG-201), has undergone testing with a limited number of patients for the prevention or reduction of keloid formation. However, intradermal injection at the site of excisional wounds yielded unsatisfactory outcomes [89]. 

Additional miRNAs and circular and long noncoding RNAs have been proposed as regulators of *COL1A1* or *COL3A1* expression. Experimentally assessed are: miR-98, miR-126- 5p, miR-218-5p, miR-328-3p, miR-338-3p [90], miR-133a and let-7 family [91]; however, none of them have been assayed as potential drugs. 

### 4.2. Prolyl-Hydroxylation

Collagens follow the secretory pathway, and numerous enzymes and chaperones play roles in collagen formation. Hydroxylation (3- and 4-) of prolines and lysins plays a crucial role in stabilizing collagens (both fibrillar and non-fibrillar) and promotes interactions with cellular collagen receptors, such as integrins and DDRs.

Prolyl-4-hydroxylation, mediated by prolyl-4-hydroxylases (*P4H*), involves α2β2 heterotetramers with catalytical α-subunits (*P4HA1*, *P4HA2*, *P4HA3*) and the structural β-subunit protein disulfide isomerase (*P4HB* or *PDI*) [92,93]. The redox activity of P4h1 requires Fe^+2^ and α-ketoglutarate. Pyridine 2,5-dicarboxylate, a mimetic of α-ketoglutarate, which inhibits P4h, notably decreased collagen production in vitro and mitigated lung fibrosis while enhancing survival in a murine model induced by bleomycin [94]. Prolyl-4-hydroxylation is a modification present in essentially all types of collagens and the drug could inhibit other enzymes that use α-ketoglutarate. Additionally, the consequences of P4h activity are not restricted to collagen; they impact the overall phenotype of the cell (e.g., stemness or Warburg effect) [95]. In this case, the treatment was locally applied via an intra-tracheal spray, initiated after injury induced by bleomycin, which improves the specificity in this complex scenario [94]. Another α-ketoglutarate analog inhibitor of P4h is ethyl-3,4-dihydroxybenzoate, which impairs the production of collagen in vitro. It is also known for its antioxidant properties [96]. An intraperitoneal treatment with this compound showed anti-metastatic effects in an orthotopic breast cancer NOD/SCID mouse model. It also somewhat slowed the growth of primary tumors, although the exact mechanism is not fully understood and could involve several pharmacological targets. Using this model, researchers found that downregulating the expression of *P4HA1* or *P4HA2* in the cancerous cells (triple negative MDA-MB-231) with specific shRNAs greatly reduced collagen levels and significantly slowed tumor progression [96]. 

Prolyl-3-hydroxylation, less common, is carried out by collagen prolyl-3-hydroxylases (*P3H1*, *P3H2*, *P3H3*, *P3H4*). The prolyl-3-hydroxylases form different complexes and modify different substrates: e.g., prolyl 3-hydroxylase 1 (*P3H1*), cartilage-associated protein (*CRTAP*), and cyclophilin B (*PPIB*) assemble into a complex within the endoplasmic reticulum that hydroxylates proline in α1(I) and α1(II) [97]. Interestingly, cyclophilin B also regulates collagen folding and contributes to both prolyl 3-hydroxylation and lysine hydroxylation of collagen [98,99]. Recently, it has been proposed that the macrocyclic natural compound sanglifehrin A targets cyclophilin B. This interaction prompts the secretion of cyclophilin B into the extracellular space, subsequently leading to a reduction in the synthesis of type I collagen. Notably, this effect occurs without inhibiting collagen mRNA transcription or inducing endoplasmic reticulum stress. The therapeutic injection of this compound exhibits a reduction in fibrosis and immune activation in both a bleomycin-induced mouse model of skin fibrosis and lung fibrosis [100]. Other potential targets include P3h2 and P3h4. P3h2 facilitates type IV collagen prolyl-hydroxylation, a process that hinders platelet aggregation in vivo [101], while P3h4 has been identified as a prognostic factor for bladder cancer and a potential target in vivo [102]. However, no drugs targeting these proteins have been reported to date.

### 4.3. Lysil-Hydroxylation

Lysyl hydroxylases, including Lh1, Lh2, and Lh3 (*PLOD1*, *PLOD2*, *PLOD3*), hydroxylate lysines within the ER, which are linked to glycosylation. While all three LH isoforms hydroxylate the helical domain. Lh1 prefers triple-helical collagen regions, while LH2 acts on noncollagenous regions (telopeptides). Lh3 also catalyzes O-glycosylation. Each isoform possesses two functionally distinct enzymatically active domains. All of this suggests that specificity could be a hallmark of Lh2 pharmacology; however, reduced activity of Lh2 produces deleterious effects in bones and its absence impedes embryogenesis [103]. Despite that, we know that *PLOD2* is upregulated and a bad prognostic factor in cancer and that Lh2 drives Epithelial-to-Mesenchymal Transition (EMT) in vitro [104]. Recently, it has been reported that Lh2-mediated extracellular matrix remodeling promotes invasiveness and metastasis in an orthotopic xenograft mice model [105]. Finally, specific inhibitors of Lh2 have been isolated and display antimigratory activity in vitro [106], although their efficacy has not yet been demonstrated in vivo. 

### 4.4. Glycosylation

The hydroxylation of lysine residues in collagen is indeed necessary for subsequent glycosylation. Collagen glycosyltransferases, Glt25d1 and Glt25d2 (*COLGALT1*, *COLGALT2*), mediate O-glycosylation of hydroxylysines with Glc(α1-2)-Gal(β1-O) or Gal(β1-O) alone. All these processes occur prior to the assembly of the three procollagen chains. This activity plays a crucial role in the stability and function of collagen fibrils in connective tissues. In this sense, the abolition or downregulation of *GLT25D1* expression leads to an accumulation of type I collagen in the ER [107], and impairs collagen deposition in a bile duct ligation-induced liver fibrosis mouse model [108]. However, no drug targeting Glt25d1 or Glt25d2 has been reported to date. 

N-glycosylation, also present in collagens, remains not fully understood. It is possibly linked to the folding process of collagens. Interestingly, it seems dispensable for collagen secretion and assembly under normal conditions. However, it could prove beneficial under conditions of ER stress [109]. N-glycosylation has not been suggested as a target for regulating collagen levels or function within the corresponding pathological microenvironment.

### 4.5. Protein Folding

Protein disulfide isomerases (PDIs) constitute a family of endoplasmic reticulum-resident proteins that play a crucial role in the folding and assembly of proteins, including collagen [110]. Specifically, PDIs assist in the formation of the triple helix structure of collagens by catalyzing the proper formation of disulfide bonds, which are essential for stabilizing collagen molecules. One of them is *P4HB* (or *PDI*), a component of the prolyl-4-hydroxylase complex described above. Interestingly, *PDI* plays a relevant role in platelet activation and thrombosis, which could be regulated with known Pdi inhibitors. However, the role of collagen in the underlying mechanism appears to be tangential [111,112]. 

The analysis of recombinant collagen I-bound material from HT-1080 clones, conducted through quantitative mass spectrometry-based proteomics, unveiled a complex interactome, which included additional members of the *PDI* family that could serve as potential targets, such as Pdia3, Pdia4, Pdia6, Erp44, and others [113]. Multiple members of the *PDI* family have been associated with cancer. For instance, *PDIA3* has been linked to cancer initiation, progression, and response to chemotherapy. It has also been identified as a potential therapeutic target in glioblastoma. Additionally, the Pdia3 inhibitor punicalagin has shown anti-glioblastoma activity in vitro [114]. However, there has been no reported evidence regarding the involvement of collagen in pathological mechanisms or the pharmacological activity of punicalagin. 

Grp78 (*HSPA5*) and Grp94 (*HSP90B1*) are highly conserved molecular chaperones that interact with collagen within the secretory pathway [115]. Grp78, a central regulator of the unfolded protein response in the endoplasmic reticulum, has also been observed to be overexpressed in cancer, notably on the outer surface of the plasma membrane (csGrp78). This localization makes it susceptible to targeting with antibodies. A monoclonal immunoglobulin M antibody (PAT-SM6) to csGrp78 had a favorable safety profile (phase 1 clinical trial) in patients with relapsed and refractory multiple myeloma [116]; however, its definitive efficacy remains elusive. Grp78 could also be targeted with organic compounds with similar purposes [117]. Likewise, Grp94 has been detected on the outer surface of the plasma membrane and found to be upregulated in cancer. Several inhibitors, including antibodies, have been tested, showing promising results [118]. In both cases (Grp78 and Grp94), the relevance of collagen expression in their mechanism of action has not been examined.

FK506-binding proteins (FKBPs) were identified based on their ability to bind tacrolimus (or FK506), an immunosuppressive drug. FKBPs are peptidyl-prolyl isomerases, catalyzing the cis-trans isomerization of peptidyl-prolyl bonds in peptides and proteins [119]. FK506-binding protein 10 (*FKBP10*) resides in the ER and functions as a chaperone for collagen I [120]. A pathogenic mutation in *FKBP10* inhibits the hydroxylation of lysine residues in the collagen telopeptide, which is essential for cross-linking, leading to osteogenesis imperfecta [121]. Downregulation of *FKBP10* expression in idiopathic pulmonary fibrosis significantly decreases the levels of collagen I and collagen V [122]. To date, there are no drugs targeting FK506-binding protein 10 that have been reported to be effective in treating collagen-mediated diseases. Heat shock protein 47 (*HSP47*) transiently binds to procollagen in the ER and dissociates in the cis-Golgi or ER-Golgi intermediate compartment region (ERGIC). It plays a central role in fibrosis, and it is a relevant target to reduce collagen expression. Increased HSP47 in cardiac fibroblasts exacerbates fibrosis and cell proliferation in ischemic hearts. And the HSP47 inhibitor Col003 inhibits HR-induced fibrogenesis in vitro [123]. One of the most advanced drugs to date is BMS-986263, a lipid nanoparticle containing HSP47 siRNA. Patients with hepatic fibrosis secondary to HCV infection were administered once-weekly intravenous infusions of BMS-986263 for 12 weeks, demonstrating good tolerability and yielding promising results [124]. 

### 4.6. Protein Trafficking

Conventional COPII-dependent vesicles are primarily involved in transporting small cargo molecules from the ER to the Golgi apparatus. However, procollagen, being a large and bulky protein, requires specialized machinery and mechanisms for its export from the ER. Tango1 (*MIA3*) and cTAGE5 (*MIA2*) are proteins involved in facilitating the ER-Golgi trafficking of collagen. They play essential roles in the formation and function of ER exit sites (ERESs), where cargo is packaged into transport vesicles for delivery [125,126].

*MIA3* knockout mice are defective in the sorting and export of several collagens, including I, II, III, IV, VII, and IX, and osteogenesis is compromised [127]. However, it is not exclusive to collagen [128]. Moreover, *MIA3* binds neurturin (*NRTN*), activating hepatocellular carcinoma cell proliferation and EMT, likely through the activation of the PI3K/AKT/mTOR pathway [129]. 

Although no drugs have been discovered to target Tango1 directly, the recent characterization of the collagen IV-Tango1 interaction represents a significant advancement [130]. This understanding paves the way for the development of drugs specifically tailored to target collagen cargo.

### 4.7. Propeptide Cleavage

Fibrillar collagens are initially secreted with intact N- and C-terminal propeptides, which must be removed for their supramolecular assembly. Enzymes crucial for propeptide cleavage include bone morphogenetic protein 1 (*BMP1*) and tolloid-like 1 and 2 (*TLL1* and *TLL2*). They belong to the astacin family of human metalloproteinases. This family targets several components of the extracellular matrix of both connective and epithelial tissues [131]. Importantly, BMP1 is not a growth factor like other BMPs, which are involved in the development and maintenance of various tissues (not just bones) [132]. 

BMP1 exhibits expression as several spliced isoforms, with antibodies targeting the BMP1-3 isoform proving effective in impairing CCl_4_-induced rat liver fibrosis [133]. Furthermore, organic compounds like UK383,367 have been shown to downregulate the maturation of procollagen I in vitro [134]. It has also demonstrated efficacy in reducing fibrosis and inflammation in a mouse model of chronic kidney disease induced by unilateral ureteral obstruction. Administration of the drug was performed intraperitoneally prior to surgery [135]. However, the relevance of the specific role of BMP1 in the pathogenesis of fibrosis has recently been questioned. This uncertainty arises from observations that BMP1 knockout mice develop fibrosis when treated with bleomycin, similar to control mice, despite BMP1 being overexpressed (in wild type) as observed in patients [136]. 

Other procollagen metalloproteinases are three members (ADAMTS2, ADAMTS3, and ADAMTS14) of the ‘a disintegrin-like and metalloprotease with thrombospondin type 1 motif’ (ADAMTS) family. This family encompasses a group of 19 members with diverse specific substrates, although not all of them have been fully characterized [137,138]. The absence of ADAMTS2 leads to the accumulation of unprocessed amino procollagen, indicating that it could be an important and accessible (extracellular) target to disrupt fibrillar collagen [137]. However, no compound targeting these enzymes has been described so far. Interestingly, several compounds targeting other members of the family have been identified, e.g., ADAMTS5, with promising results in osteoarthritis [139], suggesting that it is an option for other ADAMTS.

### 4.8. Supramolecular Assembly and Crosslinking of Fibrillar Collagens

The supramolecular assembly of collagen triple helices is a spontaneous process mediated by many interhelical interactions. The process is not fully understood, and several models have been proposed. Classical models suggest a direct cellular regulation of the process [140], with integrin-collagen interactions potentially influencing it [141]. Conversely, an alternative phase-transition model proposes that the protomers aggregate as a rapid self-assembly process because of their longitudinal structure, mechanical loading, and geometric confinement in intercellular channels [142]. The involvement of integrins or other collagen-interacting molecules in this process holds promise for potential novel pharmacological interventions in the future.

The final structure of collagen fibers relies on interprotomeric covalent bonding to stabilize the fibers, facilitated by the enzymatic deamination/oxidation of lysines. Members of the copper-dependent lysyl oxidase family (*LOX1*, *LOXL1*, *LOXL2*, *LOXL3*, and *LOXL4*) orchestrate this redox process: initially, LOX oxidatively deaminates lysine or hydroxylysine residues in collagen and elastin proteins, transforming them into allysine residues. Subsequently, two allysine residues condense in a variety of aldol dimers [143], which seems to be the major stable cross-linking bond at both ends of the type I collagen molecule in tissues that use the lysine aldehyde pathway [144]. However, it has been contested [145]. Classically, it had been stated that allysine aldols further react to produce histidinohydroxylysinonorleucine, which had been described as the main stable natural maturation product [146]. Interestingly, LOX is secreted as a proenzyme that is activated by BMP1 and ADAMTS2/14 [147].

These targets have been the most extensively assayed. In December 2010, Gilead Sciences acquired Arresto Biosciences for $225 million. Arresto Biosciences was the owner of simtuzumab (AB0024), a humanized monoclonal antibody targeting the human LOXL2. The antibody demonstrated efficacy in animal models of cancer and fibrosis [148]. Despite being subjected to 10 clinical trials and overall being well tolerated, its effectiveness remains elusive. Some of the clinical trials on AB0024 were for conditions including idiopathic pulmonary fibrosis [149], primary sclerosing cholangitis [150], primary myelofibrosis or secondary to polycythemia vera or essential thrombocythemia [151], bridging fibrosis or compensated cirrhosis caused by nonalcoholic steatohepatitis or cirrhosis due to non-alcoholic [152], liver fibrosis in HIV- and HCV-infected adults [153], steatohepatitis pancreatic adenocarcinoma (combined with gemcitabine) [154], or metastatic KRAS mutant colorectal cancer (combined with 5-fluorouracil, leucovorin, and irinotecan) [155], among others. 

The ineffectiveness of AB0024 could result from the intracellular functions of LOXL2, which are likely distinct from its lysyl oxidase activity [156]. In this scenario, small-molecule inhibitors targeting LOXL2 have undergone testing in both pre-clinical and clinical trials. Among these, PAT-1251 has been investigated the most. However, a phase II clinical trial involving the LOXL2 inhibitor PAT-1251 was halted following the acquisition of the owner company by new investors [157]. 

Alternatively, the limited effectiveness of simtuzumab could be attributed to the activity of other lysyl oxidases. To address this, broad-spectrum LOX inhibitors such as PXS-5505 have been investigated. An open-label phase 1/2a study with 39 participants indicates that PXS-5505 is safe for patients diagnosed with primary myelofibrosis, post-polycythemia vera myelofibrosis, or post-essential thrombocythemia myelofibrosis [158]. Although its efficacy has not been definitively demonstrated, preliminary results indicate promising inhibition of LOX and LOXL2 in vivo, suggesting a potential antifibrotic effect [159].

An additional clinical trial for PXS-5505 was terminated due to insufficient participation [160].

## 5. Indirect Targeting of Collagen IV: Protein Folding and Supramolecular Assembly

Each basal layer of an epithelial tissue rests on a basement membrane, which is its main component, type IV collagen.

The supramolecular assembly of type IV collagen depends on its carboxy-terminal NC1 (non-collagenous) and amino-terminal 7S domains. These domains are not proteolyzed and interact with similar domains of other protomers in the collagen IV network. The NC1 domain of two protomers interacts, forming a hexamer that is further stabilized by specific covalent bonds of sulfinimine (S=N), a cross-link of methionine and hydroxylysine residues [161]. 

The enzyme peroxidasin (*PXDN*), located within the basement membrane, generates hypohalous acid intermediates that oxidize methionine, leading to the formation of the sulfilimine cross-link. PXDN is also involved in the cross-linking of elastin. Its dysregulation has been associated with various pathological conditions [162]. The extracellular localization of PXDN, along with its relatively specific activity, renders this pharmacological approach particularly intriguing. Phloroglucinol, a known inhibitor of PXDN [163,164], emerges as a promising compound already formulated for other purposes since a daily intraperitoneal injection of phloroglucinol has been shown to reduce interstitial fibrosis in a mouse model of unilateral ureteral obstruction [165]. In vitro studies have recently identified new inhibitors of PXDN [166]. One noteworthy feature of collagen IV is its structural diversity. Differently folded NC1 variants lead to alternative assembly [14], which is potentially linked to the formation of collagen IV in the mesenchymal matrix. This mesenchymal collagen IV seems to be critical for the survival of cancerous cells undergoing EMT [15]. This alternative structure of collagen IV depends on GPBP (*COL4A3BP*) activity [15,30]. The GPBP inhibitor T12 is a terphenylic compound that mimics an interactive site, which facilitates the self-oligomerization of GPBP. Oral administration of T12 inhibits the growth of primary tumors and metastases of cancerous cells in an orthotopic and syngeneic murine model of triple-negative breast carcinoma [15]. 

## 6. Targeting with Collagen 

Animal collagen and human recombinant collagen have been extensively used during the last two decades for aesthetic and general tissue repair purposes. Synthetic nanomaterials that mimic the structure and properties of natural collagens have been developed and display hemostatic activity [167]. As these matrices are not real drugs, they fall outside of the scope of the present review and will not be addressed here. 

Whole collagen molecules are insoluble proteins that are far more difficult to use as biological drugs than antibodies. However, collagen fragments of lower molecular weight are more amenable to being used as drugs, and indeed, they have received significant attention in pharmacological studies. Enzymatic degradation of ECM proteins renders fragments named matrikines that have diverse biological activities [168]. Some of these protein fragments show exposed sites that are cryptic in the original whole polypeptides, so they are called matricryptins [169]. Different types of collagens have been identified as the original sources of matrikines and matricryptins. While some of these peptides have shown detrimental effects on health, others have been reported to be beneficial and have even been tested as potential therapeutic agents (Table 2).

### 6.1. Collagen I

One of the collagen-derived peptides with pharmacological activity is p1158/59, a 15-residue-long matricryptin produced by MMP-2/MMP-9 digestion of α1(I). Peptide p1158/59 is naturally present in human and mouse infarcted myocardium, and its synthetic counterpart has been shown to promote remodeling of left ventricle tissue and cardiac function in infarcted mouse hearts by mediating scarring and angiogenesis [170,171]. Peptide p1158/59 also potentiated the remodeling of the mouse aorta wall after induced thrombosis [172]. To our knowledge, p1158/59 has not yet entered clinical trials.

### 6.2. Collagen II

Gauci et al. [173] evidenced the existence of collagen II matrikines that promoted angiogenesis in ossification processes. The authors used a mouse knock-in approach in which the collagenase cleavage site in the α1(II) chain was mutated, thus impairing the possible generation of matrikines. The mutant mice showed abnormal ossification due to altered angiogenesis. The authors indicated their results suggested possible new approaches to enhance fracture healing, although they did not identify the pro-angiogenic collagen II matrikines whose existence they indirectly unveiled. 

### 6.3. Collagen IV

The network-forming type IV collagen is one of the main components of basement membranes, and it is subjected to matrikine-releasing proteolytical processing. Arresten and canstatin are collagen IV-derived matrikines consisting of 26-kDa α1(IV) and 24-kDa α2(IV) C-terminal fragments, respectively. Both have shown antiangiogenic activity and have been proposed as anticancer therapies [174]. Arresten and canstatin were first described in 2000 as endogenous proteolytical fragments [175,176]. Arresten has been shown to reduce the invasiveness of squamous cell carcinoma in mice [177]. Canstatin has also been shown to prevent ventricular arrhythmia in an ischemia/reperfusion rat model, partly by inhibiting the production of reactive oxygen species and the elevation of intracellular Ca^2+^ levels [178]. Additionally, canstatin also protected the rat right ventricle from fibrotic remodeling induced by experimental pulmonary arterial hypertension, possibly by interfering with Ca^2+^/calcineurin pathways [179]. However, neither arresten nor canstatin has been tested in clinical trials so far.

Type IV collagen α3 chain [α3(IV)] proteolysis by MMPs releases the 28-kDa C-terminal noncollagenous 1 (NC1) domain known as tumstatin, which displays antiangiogenic activity due to interaction with αvβ3 integrin [180]. This antiangiogenic activity was mapped to tumstatin 74–98 residues or peptide T7 [181]. T7 peptide also showed cytotoxic activity mediated, at least in part, by impairment of autophagy [182]. It should be emphasized that the tumstatin-derived T7 peptide has no relation to the so-called T7 peptide that binds to the transferrin receptor, used for liposome delivery purposes [183].

Another tumstatin fragment with proven pharmacological activity is peptide T3, (residues 69–88). T3 improved heart function and reduced heart hypertrophy, fibrosis, and oxidative stress after myocardial infarction in rats [184]. In addition, T3 inhibited apoptosis in cardiomyoblasts [185], which further underscores its utility in myocardial protection.

In relation to clinical applications with tumstatin-based drugs, in 2023 Ocugen Inc. company submitted an Investigational New Drug (IND) application to initiate a Phase 1 clinical trial with OCU200 [186], a tumstatin-transferrin fusion polypeptide. OCU200 is proposed as a treatment for diabetic macular edema, and in preclinical studies, it inhibited endothelial cell proliferation and damage in an oxygen-induced retinopathy mouse model [187].

Tetrastatin is the α4(IV)NC1 domain and was initially reported to harbor limited or no antiangiogenic properties [188]. Nevertheless, a tetrastatin-derived peptide (tetrastatin-2) evidenced antiangiogenic activity [189]. Additionally, whole tretrastatin showed antitumor properties [190], which were mapped to a 13-residue sequence known as QS-13 [191]. QS-13 itself also showed the ability to inhibit endothelial cell migration in vitro and antiangiogenic activity in mice [192]. 

The whole α5(IV)NC1 domain apparently lacked antiangiogenic activity, according to early studies [188]. However, an endogenous 20-residue-long peptide contained within the α5(IV)NC1 domain (pentastatin-1) displayed antiangiogenic properties and inhibited tumor growth in xenograft experiments in mice [193]. Pentastatin-1 also caused endothelial dysfunction and arterial pressure increase in mice, suggesting it could have a role in pulmonary hypertension [194]. Weckmann et al. found that the full α5(IV)NC1 domain, named lamstatin by the authors, inhibited lymphangiogenesis (formation of new lymph vessels) induced by tumors in mice. The activity of lamstatin was mimicked by a 17-residue peptide (CP17) encompassing amino acids 66–82 of the whole α5(IV)NC1 domain [195].

The α6(IV)NC1 domain or hexastatin has shown antiangiogenic and antitumor activity [188,196], and it has been proposed as a possible treatment for eye diseases presenting choroidal neovascularization, such as the wet form of age-related macular degeneration [197].

Despite the mentioned scientific reports, as of April 2024 no drug based on the α4(IV)NC1, α5(IV)NC1, or α6(IV)NC1 domains have yet entered clinical trials, according to the ClinicalTrials.gov database.

### 6.4. Collagen VIII

Like collagen IV, type VIII collagen is a basement membrane component that undergoes matrikine-producing degradation. In a first report, the recombinant human α1(VIII) chain C-terminal NC1 domain or vastatin inhibited proliferation and induced apoptosis of bovine aortic endothelial cells in early results, suggesting it held antiangiogenic activity [198]. Later studies found that vastatin was a human endogenous matrikine whose expression was diminished in hepatocellular carcinoma patients and that recombinant vastatin inhibited angiogenesis, tumor growth, and metastasis in a rat model of hepatocellular carcinoma [199]. Additionally, using a mouse model of glioblastoma, the antiangiogenic properties of recombinant vastatin were confirmed, and its ability to extend survival was shown [200].

### 6.5. Collagen XVIII

The basement membrane-associated type XVIII collagen is the source of endostatin, a matrikine that has received major attention in the last three decades. Endostatin is the 20-kDa C-terminal fragment of α1(XVIII) chain, and it was originally identified as an endogenous antiangiogenic polypeptide that inhibited primary tumor and metastasis growth in mice [201]. The antiangiogenic properties of endostatin rely at least on the interference with the signaling pathways of vascular endothelial growth factor (VEGF) [202], integrins [203], and nucleolin [204] in endothelial cells. Endostatin-derived drugs have been significantly used in clinical trials. As of April 2024, in the ClinicalTrials.gov database, there are 19 interventional clinical trials registered as “completed” in which recombinant human endostatin has been used as anticancer therapy. In these trials, the drug most frequently used has been Endostar™, a recombinant human endostatin expressed in *Escherichia coli* as a fusion protein with a hexahistidine tag that eases purification and increases stability. Endostar™ was approved in China in 2005 as a therapy against non-small-cell lung cancer [205]. The plasma half-life of recombinant endostatin is short, so it was conjugated to polyethylene glycol (PEG) to improve its pharmacokinetic parameters [206]. PEGylated endostatin or M2ES has been used in clinical trials as an anticancer therapy [207], but as far as we know, its approval has not yet been granted.

Endostatin-derived short synthetic peptides are antiangiogenic [208,209] and display antitumor activity [210], thus mimicking the properties of the whole α1(XVIII) fragment. Additionally, the endostatin-derived E3 peptide (residues 133–180) reduced the TGFβ-induced fibrosis in mouse skin. The C-terminal amidation of E3 rendered an even more antifibrotic peptide (E4) that, unlike other endostatin-derived peptides, lacked antiangiogenic activity [211]. The E3 analog END55, engineered to increase stability and solubility and expressed in plants, reversed established lung and skin fibrosis induced by bleomycin in mice. END55 also reduced fibrosis markers in lung explants of human patients suffering from idiopathic pulmonary fibrosis [212].

### 6.6. Collagens XV and XIX

Type XV collagen is a multidomain proteoglycan localized to basement membrane zones [213]. The α1(XV) C-terminal domain, or restin, was described as a peptide with antiangiogenic and antitumor properties [214]. However, it was later reported that while whole collagen XV and peptides derived from the N-terminal and collagenous regions displayed antitumor activity, restin failed to inhibit cervical carcinoma cell proliferation in vivo [215].

Type XIX collagen is also a component of basement membranes, and the 19-amino acid C-terminal α1(XIX)NC1 domain holds antitumor properties mediated by its ability to inhibit cancer cell invasiveness and angiogenesis [216]. Experiments with a plasmin digestion product of collagen XIX containing most of the α1(XIX)NC1 domain (peptide F4) showed the antitumor activity was due to interaction with integrins αvβ3 and α5β1 [217,218]. 

Matrikines derived from collagens XV and XIX have received far less attention than endostatin, and no clinical trials using related drugs have been registered in the ClinicalTrials.gov database so far.

## 7. Epilogue: Gene Therapy

Recently, collagen-genetic diseases have been tackled with gene therapy approaches. 

In 2023, the US FDA approved Beremagene Geperpavec for the treatment of recessive dystrophic epidermolysis bullosa (RDEB), a monogenic disease caused by mutations in *COL7A1* [219,220]. Other companies are currently developing treatments against RDEB based on recombinant expression of type VII collagen. Abeona Therapeutics has carried out clinical trials with prademagene zamikeracel (pz-cel) [221], an autologous cell therapy in which keratinocytes collected from recessive DEB patients are ex vivo engineered to express recombinant α1(VII) and transferred back to the patients. Gene delivery to patient keratinocytes is performed by transduction with a retrovirus containing the full-length α1(VII) cDNA [222,223]. Other genetic therapies are being developed against RDEB. The CRISPR/Cas9 system has been used to correct *COL7A1* frameshift mutations in fibroblasts from RDEB patients [224] and to delete mutant *COL7A1* exons in patient keratinocytes [225]. The more recently developed CRISPR/Cas9 nickase (Cas9n) system can introduce highly specific modifications in DNA while significantly avoiding off-target alterations and allowing the correction of single base substitutions, deletions, and insertions [226] The base editing and prime editing CRISPR/Cas9n systems, which rely on the expression of Cas9n fused to a base editing enzyme or to reverse transcriptase, respectively, have been used to correct several *COL7A1* gene mutations in DEB patient fibroblasts [227]. Junctional epidermolysis bullosa (JEB) is a hereditary disease that may be caused by mutations in the *COL17A1* gene, coding for type XVII collagen α1 chain [α1(XVII)] [228]. Several gene therapy-based efforts to treat JEB have been carried out. An autologous cell therapy with patient skin cells genetically modified to express recombinant α1(XVII) from a retroviral vector was used in a clinical trial [229], but the trial was terminated prior to completion. A CRISPR/Cas9-based approach was performed to correct a *COL17A1* frameshift mutation in cultured JEB keratinocytes by homology-directed repair [230]. Reframing of *COL17A1* has also been achieved by CRISPR/Cas9 nickase-based paired nicking [231]. However, although the CRISPR/Cas9 nickase system has reduced the rate of off-target alterations in the original CRISPR/Cas9 nuclease system, the risk of the appearance of unwanted genomic alterations persists [232].

Finally, gene therapy treatments against osteogenesis imperfecta (OI) caused by mutations in type I collagen genes are also being developed. Working with an OI mouse model and using adeno-associated virus (AAV) for delivery purposes, Yang et al. carried out a CRISPR/Cas9, homology-directed repair approach to correct OI-causing *Col1a2* mutations in vivo, and they reported an amelioration of the animal phenotypes [233].

## 8. Conclusions

The overall effectiveness of targeting collagen and of targeting with collagen-based therapies is currently limited (Table 3). Only a few drugs have shown success in clinical trials, and their applications are restricted to a limited number of diseases. A key challenge when targeting collagen-related pathways for a particular pathological condition is specificity. However, ongoing research focused on understanding specific collagen isoforms, their interactions, and the underlying molecular mechanisms of collagen-related diseases holds promise. By gaining a deeper understanding of these factors, it may be possible to develop more precise therapeutic interventions. In this sense, a disease-specific diversification of collagen I has been described [234]. And recombinant collagen IV exhibits a similar phenomenon of structural diversification [16]. It could pave the way for monoclonal antibodies designed to target these disease-specific collagen formations. These targeted treatments have the potential to offer improved efficacy and fewer adverse effects, addressing the current limitations of collagen pharmacology.

## Figures and Tables

**Figure 1 ijms-25-06523-f001:**
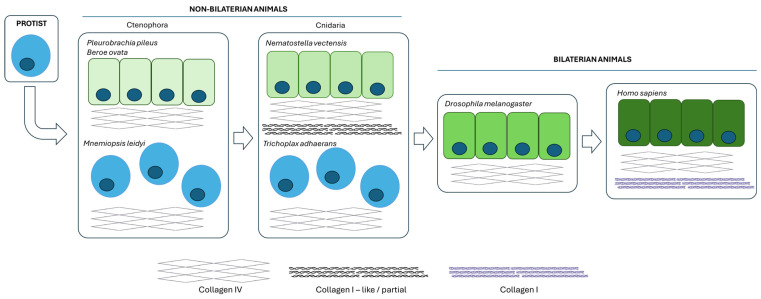
Phylogeny of collagen. Collagen IV is the earliest form of collagen and is normally associated with epithelial differentiation. Collagen I emerged in more advanced organisms and serves as a key component of the mesenchymal extracellular matrix. This illustration is inspired by the findings, schemes, and conclusions of Billy G. Hudson’s laboratory [12].

**Figure 2 ijms-25-06523-f002:**
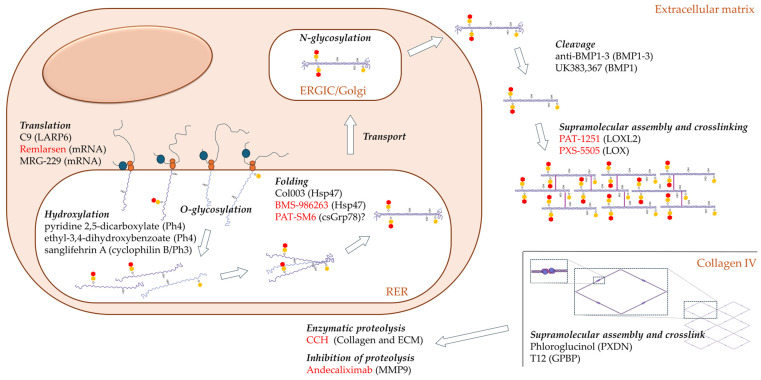
Direct and indirect targeting of collagen. Schematic representation of relevant cellular processes and drugs targeting collagen expression, cleavage, assembly, and degradation in the extracellular matrix. The different targetable processes are indicated with cursive bold lettering, and the drugs listed below with the corresponding target in parentheses. In red are the drugs assayed in clinical trials registered in ClinicalTrials.org. A question mark (?) denotes that a not fully demonstrated drug target.

**Table 1 ijms-25-06523-t001:** Human collagens: types, protomeric combinations, and dysfunctional phenotypes described in OMIM. Inspired by a table of Sylvie Ricard-Blum [1]. ^a^ In red: proteoglycans (contain glycosaminoglycans). ^b^ Non-pathological alternative splicing variants. In red: primary autoantigen in autoimmune diseases. ^c^ A pseudogene in humans with two variants is denoted with italics. ^d^ It has been also described as α1(XXIX). ^e^ α1(II) has been also described as α3(XI). Chains from different collagen types are highlighted. Question marks (?) are used to denote triple helical combinations not fully characterized or phenotypes whose relationship with the indicated collagen chains is not fully demonstrated.

Organization	Type (+GAG) ^a^	Chains (with alt. Splicing Isoforms; Autoantigen) ^b^	Triple Helical Combinations	Phenotype of Human Mutations (Affected Chain)
**Basement membrane network**	IV	α1(IV) α2(IV) α3(IV) α4(IV) α5(IV) α6(IV)	[α1(IV)]_2_ α2(IV)	Angiopathies, nephropathy (α1, α2). Kidney disease, hematuria, loss hearing, eye abnormalities (α3, α4, α5). Deafness (α6)?.
			α3(IV) α4(IV) α5(IV)	
			[α5(IV)]_2_ α6(IV)	
**Basketweave like network**	VI	α1(VI) α2(VI) α3(VI) *α4(VI)* ^c^ α5(VI) ^d^ α6(IV)	α1(VI) α2(VI) α3(VI)	Muscular dystrophy, myopathy (α1, α2, α3). Involuntary movements, dystonia (α3). Chronic neuropathic itch (α5)?
			α1(VI) α2(VI) α5(VI) ?	
			α1(VI) α2(VI) α6(VI) ?	
**Hexagonal networks**	VIII	α1(VIII) α2(VIII)	[α1(VIII)]_2_ α2(VIII)	Corneal dystrophy (α2).
			α1(VIII) [α2(VIII)]_2_	
			[α1(VIII)]_3_	
			[α2(VIII)]_3_	
	X	α1(X)	[α1(X)]_3_	Chondrodysplasia.
**Fibrillar**	I	α1(I) α2(I)	[α1(I)]_2_ α2(I)	Aberrant osteogenesis, osteoporosis, overly flexible joints, stretchy-fragile skin (α1, α2).
			[α1(I)]_3_	
	II	α1(II)	[α1(II)]_3_	Hipochondrogenosis, spondyloepiphyseal dysplasia, retinal detachment.
	III	α1(III)	[α1(III)]_3_	Joint laxity and stretchy-fragile skin, vascular problems and aortic dissection.
	V	α1(V) α2(V) α3(V)	[α1(V)]_2_ α2(V)	Corneal problems, fibromuscular dysplasia, skin hyperextensibility, dystrophic scarring, and joint hypermobility (α1, α2).
			[α1(V)]_3_	
			α1(XI) α1(V) α1(II) ^e^	
	XI	α1(XI) α2(XI)	α1(XI) α2(XI) α1(II) ^e^	Ophthalmologic, deafness, skeletal abnormalities, fibrochondrogenesis (α1, α2).
			α1(XI) α1(V) α1(II) ^e^	
	XXIV	α1(XXIV)	[α1(XXIV)]_3_	
	XXVII	α1(XXVII)	[α1(XXVII)]_3_	Short stature, bilateral congenital hip dislocation, radial head dislocation, carpal coalition, scoliosis, dysmorphic face.
**Fibrillar-associated collagens with interrupted triple helices (FACIT)**	IX	α1(IX) α2(IX) α3(IX)	α1(IX) α2(IX) α3(IX)	Epiphyseal dysplasia, arthro-ophthalmodystrophy (α1, α2, α3).
	XII	α1(XII)	[α1(XII)]_3_	Muscular dystrophy, myopathy.
	XIV	α1(XIV)	[α1(XIV)]_3_	Punctate abnormal thickening of the stratum corneum of the palms and soles?
	XVI	α1(XVI)	[α1(XVI)]_3_	
	XIX	α1(XIX)	[α1(XIX)]_3_	
	XX	α1(XX)	[α1(XX)]_3_	Striate abnormal thickening of the stratum corneum of the palms and soles?
	XXI	α1(XXI)	[α1(XXI)]_3_	
	XXII	α1(XXII)	[α1(XXII)]_3_	
**Anchoring ** **fibrils**	VII	α1(VII)	[α1(VII)]_3_	Cutaneous and mucosal fragility resulting in blisters and superficial ulcerations.
**Membrane bound**	XIII	α1(XIII)	[α1(XIII)]_3_	Skeletal muscle weakness.
	XVII	α1(XVII)	[α1(XVII)]_3_	Atrophy of the skin and nonscarring blistering, epithelial dystrophy.
	XXIII	α1(XXIII)	[α1(XXIII)]_3_	
	XXV	α1(XXV)	[α1(XXV)]_3_	Fibrosis of extraocular muscles, ophthalmoplegia.
**Multiplexins**	XV	α1(XV)	[α1(XV)]_3_	
	XVIII	α1(XVIII)	[α1(XVIII)]_3_	High myopia, vitreoretinal degeneration and occipital skull defects
**Other**	XXVI	α1(XXVI)	[α1(XXVI)]_3_	
	XXVIII	α1(XXVIII)	[α1(XXVIII)]_3_	

**Table 2 ijms-25-06523-t002:** Targeting with collagen. Matrikines and matricryptins: origin and pharmacological activities.

Collagen	Chain	Matrikine/Matricryptin Registered in ClinicalTrials.org	Activity
I	α1	p1158/59	Potentiates the remodeling of aorta wall after thrombosis
IV	α1	Arresten	Antiangiogenic
	α2	Canstatin	Antiangiogenic. Inhibits the production of ROS and the elevation of intracellular Ca^2+^ levels. Antifibrotic
	α3	Tumstatin	Antiangiogenic
		Tumstatin-peptide T7	Cytotoxic
		Tumstatin-peptide T3	Improves heart function and reduces heart hypertrophy, fibrosis, and oxidative stress after myocardial infarction
		Tumstatin-transferrin fusion polypeptide: OCU200	Inhibits endothelial cell proliferation and damage in an oxygen-induced retinopathy
	α4	Tetrastatin-2	Antiangiogenic
		Tetrastatin peptide QS-13	Antiangiogenic. Antimigratory
	α5	Pentastatin-1	Antiangiogenic. Antitumoral
		Lamstatin	Inhibits lymphangiogenesis
		Lamstatin peptide CP17	
	α6	Hexastatin	Antiangiogenic. Antitumoral
VIII	α1	Vastatin	Antiangiogenic. Antitumoral
XV	α1	Restin	Antiangiogenic. Antitumoral
XVIII	α1	Endostatin	Antiangiogenic. Antitumoral
		PEGylated endostatin M2ES	
		Endostar™ (recombinant endostatin approved in China)	
		Endostatin-derived E3 peptide	Antifibrotic
		E3 analogue END55	
XIX	α1	Peptide F4	Antiangiogenic. Antimetastatic

**Table 3 ijms-25-06523-t003:** Pharmacology of collagen: treatments assayed in clinical trials.

Group	Drug (Type)	Clinical Trial	Disease	Refs
Direct	Collagenase from *Clostridium histolyticum*	Marketed	Fibrosis: Dupuytren’s contracture	[42,43,44]
			Fibrosis: Peyronie’s disease	[45]
			Debriding chronic dermal ulcers and severely burned areas	[51,52]
		Phase II: efficacy not proven	Adhesive capsulitis of the shoulder	[46]
Indirect	MMPs inhibitors (12)	Phase I–III: efficacy not proven	Cancer	[55,56]
			Macular degeneration	
			Arthritis	
			Inflammation	
	MMP-9 inhibitor	Phase II and III: efficacy not proven	Gastric Cancer, Advanced Gastric, or GEJ Adenocarcinoma	[57,58]
		Phase II: efficacy not proven	Crohn’s Disease	[59]
	mimic of miR-29	Phase II: efficacy not proven	Keloid formation	[88,89]
	anti-Grp78	Phase I: favorable safety profile	Refractory multiple myeloma	[116]
	HSP47 siRNA	Phase II: promising results	Hepatic fibrosis secondary to HCV infection	[124]
	Anti-LOXL2	Phase II: efficacy not proven	Idiopathic pulmonary fibrosis	[149]
			Primary sclerosing cholangitis	[150]
			Primary myelofibrosis	[151]
			Myelofibrosis secondary to polycythemia vera	[151]
			Essential thrombocythemia	[151]
			Bridging Fibrosis or Compensated Cirrhosis Caused by Nonalcoholic Steatohepatitis	[152]
			Liver fibrosis in HIV and HCV-infected adults	[153]
			Steatohepatitis pancreatic adenocarcinoma	[154]
			Metastatic KRAS mutant colorectal cancer	[155]
	Broad-spectrum LOX inhibitors	Phase I/II: promising results	Myelofibrosis	[159]
Collagen Peptides	Tumstatin based drugs [α3(IV)]	Phase I:	Diabetic macular edema	[186]
	Endostatin, PEGylated endostatin [α1(XVIII)]	Phase III—marketed	Cancer	[205]
Gene therapy	Recombinant *COL7A1* expression	Marketed	Dystrophic epidermolysis bullosa	[219,220]

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
