# Peer review of "The Versatility of Collagen in Pharmacology: Targeting Collagen, Targeting with Collagen"

_ijms, 2024, doi:10.3390/ijms25126523_

Round 1

Reviewer 1 Report

Comments and Suggestions for Authors

Review note

This paper reviews the collagen targeting and the application of collagen and its derivatives in the development of novel treatments for a range of pathological conditions. This topic is importangt. I appreciate the authors’ works, which will help further basic research and (pre)clinical practice. However, to grow into a publication, I think there are some issues the authors need to address.

1.     In section 2, the authors randomly pick up several 3 (types) diseases and discussed their relations with collagen. Indeed, this section is not comprehensive as it does not align with the primary objective of this review. I would suggest the authors to draw this section in a more generalized way that try to cover the entire spectrum of collage related diseases. 

2.     The authors used direct targeting and indirect targeting to categorize the application of collagen and its derivatives in development of new treatments. Although the specific targeted diseases have been integreated into each section, it is still worth to recap the diseases using a single section, through which the targeted diseases are listed one after another. 

3.     More schematics are encouraged for a review paper.

In summary, I feel the review is solid. However, minor revisions are needed. I hope the author(s) could find some of the above discussions helpful for improving the paper. 

Comments on the Quality of English Language

Minor editing of English language required

Author Response

First, thank you for your interest and for analyzing our proposal. We have made changes to incorporate all the suggestions provided by the reviewers:

Following your suggestion, we have included a new table with the treatments and the corresponding diseases. It is referenced in the conclusions. Additionally, we have added the genetic diseases mentioned in section 7. We aimed to list these diseases with minimal descriptions briefly.

A list of the major modifications:

The title: It now reads: “The Versatility of Collagen in Pharmacology: Targeting Collagen, Targeting with Collagen.” Initially, we aimed for a simpler, more informative title.

References for collagen genes: Listing all collagen genes would require numerous references. However, a good alternative is the OMIM database. We have included an additional Table S1 encompassing all the collagen genes (COLs) with the corresponding relevant information found in OMIM. This paragraph concludes: “Enclosed, please find a comprehensive list of collagen genes in the OMIM database (Table S1).”

Section 1: Added a cartoon/scheme of the phylogeny of collagen, based on descriptions and analysis from Billy G. Hudson’s laboratory.

Section 2: a. Introduced pancreatic fibrosis. b. To improve this complex section we have included a mention of the genetic diseases discussed in the section “7. Epilogue: Gene Therapy.”

Section 3: a. All ClinicalTrials references are now included.

Table 1: We have included the reference that inspired us.

Conclusions: We have summarized the list of diseases with treatments and results from clinical trials in a new Table (Table 3).

Reviewer 2 Report

Comments and Suggestions for Authors

Overall, an excellent and comprehensive review. Minor edits required. Please find attached some comments for consideration. 

Comments on the Quality of English Language

Requires changes/modifications 

Author Response

First, thank you for your interest, the detailed analysis of the paper, and the kind words you have written about our proposal. Your suggestions have improved it considerably. We have accepted almost all your suggestions, except for two:

The title: We incorporated 50% of your suggestion. It now reads: “The Versatility of Collagen in Pharmacology: Targeting Collagen, Targeting with Collagen”. Initially, we aimed for a simpler, but informative title, maybe too short.

References for collagen genes: Listing all collagen genes would require numerous references. However, a good alternative is the OMIM database. We have included an additional Table S1 encompassing all the collagen genes (COLs) with the corresponding relevant information found in OMIM. This paragraph concludes: “Enclosed, please find a comprehensive list of collagen genes in the OMIM database (Table S1).”

Major modifications were:

Section 1: Added a cartoon/scheme of the phylogeny of collagen, based on descriptions and analysis from Billy G. Hudson’s laboratory.

Section 2: a. Introduced pancreatic fibrosis. b. To improve this complex section, and following another reviewer’s recommendation, we have included a mention of the genetic diseases discussed in section “7. Epilogue: Gene Therapy.”

Section 3: a. All ClinicalTrials references are now included. b. Figure 2 could be seen as a flowchart.

Table 1: We have included the reference that inspired us.

Conclusions: We have summarized the list of diseases with treatments and results from clinical trials in a new Table (Table 3).